# Natural Activators of Autophagy Increase Maximal Walking Distance and Reduce Oxidative Stress in Patients with Peripheral Artery Disease: A Pilot Study

**DOI:** 10.3390/antiox11091836

**Published:** 2022-09-18

**Authors:** Ombretta Martinelli, Mariangela Peruzzi, Simona Bartimoccia, Alessandra D’Amico, Simona Marchitti, Speranza Rubattu, Giovanni Alfonso Chiariello, Luca D’Ambrosio, Sonia Schiavon, Fabio Miraldi, Wael Saade, Mizar D’Abramo, Annachiara Pingitore, Lorenzo Loffredo, Cristina Nocella, Maurizio Forte, Pasquale Pignatelli

**Affiliations:** 1Department of General and Specialistic Surgery “Paride Stefanini”, Sapienza University of Rome, 00161 Rome, Italy; 2Department of Clinical Internal, Anesthesiology and Cardiovascular Sciences, Sapienza University of Rome, 00161 Rome, Italy; 3Mediterranea Cardiocentro, 80122 Naples, Italy; 4Department of Movement, Human and Health Sciences, University of Rome “Foro Italico”, 00135 Rome, Italy; 5IRCCS Neuromed, Località Camerelle, 86077 Pozzilli, Italy; 6Cardiology Unit, Department of Clinical and Molecular Medicine, School of Medicine and Psychology, Sant’Andrea Hospital, Sapienza University of Rome, 00189 Rome, Italy; 7Cardiovascular Sciences Department, Agostino Gemelli Foundation Polyclinic IRCCS, Catholic University of the Sacred Heart, 00168 Rome, Italy; 8Department of Medical-Surgical Sciences and Biotechnologies, Sapienza University of Rome, 04100 Latina, Italy

**Keywords:** autophagy, oxidative stress, peripheral artery disease, trehalose, polyphenols

## Abstract

Trehalose, spermidine, nicotinamide, and polyphenols have been shown to display pro-autophagic and antioxidant properties, eventually reducing cardiovascular and ischemic complications. This study aimed to investigate whether a mixture of these components improves maximal walking distance (MWD) in peripheral artery disease (PAD) patients. Nitrite/nitrate (NOx), endothelin-1, sNOX2-dp, H_2_O_2_ production, H_2_O_2_ break-down activity (HBA), ATG5 and P62 levels, flow-mediated dilation (FMD), and MWD were evaluated in 20 PAD patients randomly allocated to 10 g of mixture or no-treatment in a single-blind study. The above variables were assessed at baseline and 60 days after mixture ingestion. Compared with baseline, mixture intake significantly increased MWD (+91%; *p* < 0.01) and serum NOx (+96%; *p* < 0.001), whereas it significantly reduced endothelin-1 levels (−30%, *p* < 0.01). Moreover, mixture intake led to a remarkable reduction in sNOX2dp (−31%, *p* < 0.05) and H_2_O_2_ (−40%, *p* < 0.001) and potentiated antioxidant power (+110%, *p* < 0.001). Finally, mixture ingestion restored autophagy by increasing ATG5 (+43%, *p* < 0.01) and decreasing P62 (−29%, *p* < 0.05). No changes in the above-mentioned variables were observed in the no-treatment group. The treatment with a mixture of trehalose, spermidine, nicotinamide, and polyphenols improves MWD in PAD patients, with a mechanism possibly related to NOX2-mediated oxidative stress downregulation and autophagic flux upregulation. Clinical Trial Registration unique identifier: NCT04061070.

## 1. Introduction

Peripheral arterial disease (PAD) is a chronic disorder resulting from the presence of arterial obstructive disease, which causes inadequate blood flow to both upper and lower extremities, most frequently affecting the lower limbs [1]. In most cases, the underlying disease process is atherosclerosis, which primarily affects the arteries of the lower limbs, and is often a multi-district vascular disease involving the coronary circulation as well as the cerebral arteries. In fact, epidemiological studies have shown that up to 50% of patients with PAD also have cerebrovascular or cardiac symptoms [2]. Moreover, both traditional cardiovascular disease risk factors, such as current smoking, diabetes, hypertension, and hypercholesterolemia, as well as non-traditional risk factors such as fibrinogen and C-reactive protein, were positively associated with PAD, reflecting the high burden of atherosclerosis in these patients [3].

Intermittent claudication, defined as a reduction in pain-free walking distance, is the typical clinical manifestation of the disease. It affects one-third of PAD patients and is identified by impaired blood flow to the limbs during exercise. Intermittent claudication remains stable over 5 years in 70–80%, worsens in 10–20%, and evolves into critical limb ischemia in 5–10% of patients [2]. Deranged endothelial function, accumulation of toxic metabolites, alteration in nitric oxide (NO) generation, and increased oxidative stress play a synergistic role among the flow-reducing factors in PAD patients.

Autophagy is a cytoprotective intracellular process that mediates degradation of proteins, turnover of organelles, and recycling of cytoplasmic components in conditions of nutrient deprivation and cellular stress [4]. Furthermore, autophagy plays an important role in the removal of excessive cellular ROS while maintaining a redox balance [5]. As the activation of autophagic flux favours cell adaptation to energetic and stressful changes by supporting cell metabolism, homeostasis, and survival [6], insufficient autophagic activity can contribute to the pathogenesis of cardiovascular diseases, diabetes, inflammatory disorders, cancer, and physical stress [7]. 

Natural activators of autophagy were observed to exert beneficial effects in pre-clinical models of cardiovascular diseases, acting on specific molecular targets, such as the mechanistic target of rapamycin complex I (mTORC1), transcription factor EB (TFEB), and 5′ adenosine monophosphate-activated protein kinase [8,9]. Trehalose is a natural disaccharide composed of two glucose molecules linked together by an α1-1-glycosidic bond, synthesized by lower organisms such as yeasts, bacteria, insects, and plants, but not by mammals. Trehalose performs multiple functions including a protective action against various stressors, such as oxidative stress, temperature excursions, accumulation of protein aggregates, and dehydration [10]. Furthermore, recent evidence showed that trehalose could prevent inflammatory responses induced by endotoxic shock both in vivo and in vitro [11,12]. Moreover, trehalose was previously shown to be a strong inducer of autophagy, and previous work demonstrated that its oral administration can drastically reduce the development and progression of neurodegenerative disorders, hepatic steatosis, insulin resistance, post-ischemic cardiac remodelling, atherosclerosis, and stroke [13,14,15,16,17,18]. These beneficial effects are partially mediated by the stimulation of autophagy, as they are attenuated in models of autophagy inhibition. Accumulating lines of evidence also demonstrated that spermidine, polyphenols, and nicotinamide are potent natural activators of autophagy. Spermidine is an endogenous amine that has been reported to extend lifespan in mice and to reduce cardiovascular aging and atherosclerosis through an autophagy-dependent mechanism [19,20]. Nicotinamide was also reported to reactivate autophagy in a model of spontaneous stroke and to improve mitochondrial quality control in human cells [21,22]. It was previously shown that trehalose in combination with a mixture of spermidine, nicotinamide, and polyphenols (catechin and epicatechin) was able to reduce platelet activation and oxidative stress and to increase the production of nitric oxide, angiogenesis, and cell viability in platelets isolated from patients with atrial fibrillation, metabolic syndrome, or smokers [23]. 

As trehalose, spermidine, nicotinamide, and polyphenols could reduce the impact of risk factors on cardiovascular and ischemic complications [9,23], the purpose of this study was to evaluate if the administration of a mixture of natural activators of autophagy is able to down-regulate ROS production and ameliorate maximal walking distance (MWD) in PAD patients by improving autophagic function and oxidative stress. 

## 2. Materials and Methods

An interventional trial was performed by recruiting PAD patients to investigate the chronic effect (2 months) of a mixture of natural activators of autophagy (10.5 g/twice day) on maximal walking distance (MWD), flow-mediated dilation (FMD), and oxidative stress, as assessed by blood levels of soluble NOX2-derivative peptide (sNOX2-dp), a marker of NOX2 activation, H_2_O_2_ production, and HBA and NO generation, as assessed by serum levels of nitrite/nitrate (NOx). Moreover, markers of autophagy, such ATG5, and P62 were assessed in plasma.

Twenty PAD patients in Fontaine stage IIb agreed to participate to the study, which was performed between March 2021 and May 2022. The recruited patients had been suffering from PAD for 18–24 months (mean 8 months). Patients had to be in stable conditions, without abrupt changes of walking distance and ABI in the month before the study.

They were randomly allocated to a treatment sequence with mixture or no-treatment in a single-blind design. The treatment mixture was administered daily for 2 months. Flow-mediated dilatation (FMD) and MWD assessed by 6 min walking test, oxidative stress, and autophagy parameters were assessed at baseline (T0) and after 2 months of treatment (T2M).

A full medical history and physical examination was recorded for all participants. Exclusion criteria were liver failure, severe kidney disorders (serum creatinine (Mt) 2.8 mg/dL), acute cerebrovascular disease, acute myocardial infarction, current smoking, or treatment with antioxidants. No dropouts or missing data were observed during the study.

The study protocol was approved by the local ethical board of Sapienza—University of Rome (ClinicalTrials.gov Identifier: NCT04061070) and was conducted according to principles of the 1975 Declaration of Helsinki. All patients provided written informed consent at baseline.

### 2.1. Mixture Composition

The mixture was obtained from Princeps srl (Piasco, Cuneo, Italy). The mixture composition is reported in Table 1.

### 2.2. Study Outcomes

The primary outcome of this study was to evaluate the effect of the mixture administration on walking distance in PAD patients.

### 2.3. Randomization and Blinding

The study treatment codes were assigned by an individual not involved in the study that randomly assigned participants to a treatment sequence with mixture or no-treatment and kept the key in a sealed envelope. The randomization was computer-generated based on a random numeric sequence. The authors and laboratory technicians were unaware of the treatment allocation.

### 2.4. Blood Sampling and Preparations

For each patient, blood samples were collected in the morning (8–9 a.m.) from the antecubital vein in a seated position in fasting conditions. Blood samples were collected in BD Vacutainer (Franklin Lakes, NJ, USA) without anticoagulants or with anticoagulants (trisodium citrate, 3.8%, 1/10 (*v*/*v*) or 7.2 EDTA). The blood was centrifuged at 300× *g* for 10 min at room temperature (RT). Serum and plasma samples were separated into aliquots and stored at −80 °C until analyses.

### 2.5. Six Min Walking Test

Patients performed the 6 min walking test, which was administered by trained exercise technicians. Two cones were placed thirty meters apart in a marked corridor; technicians supervised the test and instructed the patients to walk as many laps around the cones as possible for up to 6 min. During the test, patients indicated if and when they experienced the onset of claudication pain. The pain-free total distance walked during the test was recorded [24].

### 2.6. Ankle–Brachial Index Measurements

Ankle–brachial index (ABI) was performed as previously described [25,26]. 

### 2.7. FMD Measurement

Ultrasound assessment of FMD was investigated according to the guidelines as previously reported [27]. The coefficient of variation for FMD measurements, obtained on three separate occasions, was 12.5%.

Longitudinal ultrasonographic scans of the carotid artery were obtained on the same day as the studies of the brachial artery reactivity and included the evaluation of the right and left common carotid arteries 1 cm proximal to the carotid bulb. FMD was performed with a 7.5 MHz linear-array transducer ultrasound system (Sonomed, Lake Success, NY, USA). 

### 2.8. Serum Nitric Oxide

Nitric oxide (NO) was evaluated in serum by NO^2−^/NO^3−^ determination. Briefly, the nitrate (NO^3−^) in the sample is converted into nitrite (NO^2−^) by nitrate reductase enzyme, and then total nitrite is detected with Griess Reagents as a coloured azo dye product (absorbance 540 nm). Values were expressed as μM. Intra- and inter-assay coefficients of variation were <10%.

### 2.9. Serum Endothelin 1 Measurements

Quantitative determination of human endothelin 1 was performed by commercial ELISA kit (Elabscience). Values were expressed as pg/mL. Intra- and inter-assay coefficients of variation (CVs) were both <10%.

### 2.10. Serum sNOX2-dp

NOX2 activity was measured in serum as sNOX2-dp with a previously reported ELISA method [28]. Values were expressed as pg/mL; intra- and inter-assay coefficients of variation (CVs) were <10%.

### 2.11. Serum H_2_O_2_ Production

Hydrogen peroxide (H_2_O_2_) was measured by a colorimetric assay as described previously [29]. The final product was read at 450 nm and expressed as μM. Intra- and inter-assay CVs were both <10%.

### 2.12. Serum Hydrogen Peroxide Scavenging Activity 

Hydrogen peroxide (H_2_O_2_) break-down activity by HBA assay kit (Aurogene, code HPSA-50) was measured to evaluate the antioxidant capacity of serum samples. The % of HBA was calculated according to the following formula: % of HBA = [(Ac − As)/Ac] × 100, where Ac is the absorbance of H_2_O_2_ 1.4 mg/mL and As is the absorbance in the presence of the serum sample.

### 2.13. Plasma ATG5 Detection 

Autophagy protein 5 (ATG5), belonging to the ATG family, is an essential protein in the process of autophagy. ATG5 is required in the early stages of the autophagosome formation, a double-membrane vesicle that delivers cytoplasmic material to the lytic compartment of the cell for degradation [30].

Quantitative determination of ATG5 in plasma samples was performed by the sandwich enzyme immunoassay technique (Mybiosource, No. MBS2602759). The sample concentration was determined using a microplate reader set to 450 nm and values were expressed as ng/mL. Intra-assay and inter-assay coefficients of variation were ≤8% and ≤12%, respectively.

### 2.14. Plasma P62 Detection

The P62 is a ubiquitin-binding scaffold protein that enables ubiquitinated protein degradation in the lysosome by linking them to the autophagic machinery. P62 may be used as a marker of autophagic flux as it is degraded by autophagy and accumulates when autophagy is inhibited [31]. 

Plasmatic P62 was analysed by a sandwich enzyme immunoassay technology (FineTest, No. EH10842). The concentration of the protein can be calculated by reading the O.D. absorbance at 450 nm. Values were expressed as ng/mL. Intra-assay and inter-assay coefficients of variation were <8 and <10%, respectively.

### 2.15. Sample Size Calculation

Given the pilot nature of the study and the absence of preliminary data on this issue, a sample size was not formally calculated.

### 2.16. Statistical Analyses

Continuous variables are reported as mean ± SD. Comparisons between groups were conducted by Student’s *t*-test. The assessment of treatment and period effects was carried out by performing a split-plot ANOVA with one between-subject factor (treatment sequence) and two within-subject factors (period 1 versus 2; pre- versus post-treatment). Bivariate analysis was performed by Spearman rank correlation test. A value of *p* < 0.05 was considered statistically significant. All tests were performed using GraphPad Software-Prism7. 

## 3. Results

Clinical characteristics of PAD patients are reported in Table 2. 

### 3.1. Clinical Outcome

After 2 months, a significant difference for treatments (mixture vs. no-treatment) was found with respect to MWD (*** *p* < 0.001; Figure 1A). Concerning FMD, we found an ameliorative trend in PAD patients after treatment with the mixture even if it did not reach statistical significance (Figure 1B). Finally, no changes were observed in ABI measurement (Figure 1C).

Compared with the baseline, MWD increased significantly after mixture intake (from 152.7 ± 115.5 to 282 ± 138.5 m, ** *p* < 0.01); no changes were observed after no treatment (from 112.1 ± 32.5 to 103 ± 27.8 m, *p* = 0.515 for MWD) (Figure 1A). Conversely, FMD showed an improvement after mixture intake even if it did not reach statistical significance (from 1.82 ± 0.5 to 2.33 ± 0.5%, *p* = 0.492) (Figure 1B). In addition, a non-significant worsening trend was observed after no treatment (from 1.42 ± 0.8 to 0.93 ± 0.4%, *p* = 0.592) (Figure 1B). No significant effect was found after mixture intake or no treatment for ABI (from 0.73 ± 0.23 to 0.76 ± 0.19, *p* = 0.372 and from 0.69 ± 0.33 to 0.71 ± 0.34, *p* = 0.999, respectively) (Figure 1C) and for systolic pressure (from 149.0 ± 9.4 to 141.5 ± 9.4 mmHg, *p* = 0.148 and from 150.5 ± 9.8 to 149.0 ± 4.6 mmHg, *p* = 0.963, respectively). Conversely, we found a significant decrease in dyastolic pressure after mixture intake (from 84.5 ± 6.0 mmHg to 78.0 ± 6.3 mmHg, *p* = 0.03) and no changes after no treatment (from 84.5 ± 5.5 to 82.0 ± 5.9, *p* = 0.483).

### 3.2. Endothelial Dysfunction

After 2 months, a significant difference for treatments (mixture vs. no-treatment) was found with respect to NO production (*** *p* < 0.001; Figure 1D) and endothelin concentration (* *p* < 0.05; Figure 1E). Compared with the baseline, NO production was potentiated after mixture intake (from 32.41 ± 1.7 to 63.6 ± 2.26 μM, *** *p* < 0.001), whereas endothelin-1 concentration was significantly reduced (from 21.97 ± 1.47 to 15.33 ± 1.24 pg/mL, ** *p* < 0.01) after mixture intake. No changes were observed after no treatment (from 32.51 ± 2.36 to 40.08 ± 3.80 μM, *p* = 0.173 for NO and from 18.97 ± 1.37 to 20.83 ± 1.38 pg/mL, *p* = 0.353 for endothelin-1) (Figure 1D,E). 

### 3.3. Oxidative Stress Evaluation

A significant difference for treatments (mixture vs no treatment) was found with respect to sNOX2-dp release (** *p* < 0.01; Figure 2A), H_2_O_2_ (*** *p* < 0.001; Figure 2B), and HBA (*** *p* < 0.001; Figure 2C). The pairwise comparisons showed that sNOX2-dp, and H_2_O_2_ significantly decreased after mixture intake (from 32.26 ± 1.50 to 22.22 ± 1.58 pg/mL, * *p* < 0.05 and from 26.1 ± 1.35 to 15.5 ± 1.43 μM, *** *p* < 0.001, respectively), while no changes were observed after no treatment (from 42.0 ± 7.6 to 45.5 ± 6.7 pg/mL, *p* = 0.757 and from 24.16 ± 1.5 to 25.7 ± 2.3 μM, *p* = 0.581, respectively) (Figure 2A,B). Conversely, HBA, an index of the antioxidant power, significantly increased after mixture intake (from 16.78 ± 0.98 to 35.29 ± 3.75%, *** *p* < 0.001). No changes were observed after no treatment (from 16.25 ± 1.14 to 19.52 ± 2.83%, *p* = 0.297) (Figure 2C). 

### 3.4. Autophagy Evaluation

To explore the role of autophagy, we evaluated the levels of P62 and ATG5 in the plasma of PAD patients. This analysis revealed that, compared with control patients, mixture administration significantly increased the levels of ATG5 protein (** *p* < 0.01; Figure 3A) and decreased serum P62 levels (** *p* < 0.01; Figure 3B). Compared with the baseline, serum levels of ATG5 significantly increased (from 93.52 ± 3.06 to 133.8 ± 5.7 ng/mL, ** *p* < 0.001) and P62 significantly decreased (from 75.64 ± 6.68 to 53.28 ± 6.24 ng/mL, * *p* < 0.05), while no changes were observed in control patients (from 88.67 ± 10.65 to 95.12 ± 11.8 ng/mL, *p* = 0.690 for ATG5 and from 83.93 ± 6.46 to 88.67 ± 7.04, *p* = 0.724 for P62). These results suggest that autophagy increases in subjects receiving the mixture.

### 3.5. Linear Correlation

A linear correlation analysis showed that ∆ (expressed by difference of values between before and after mixture intake) of MWD correlated with ∆ of sNOX2-dp (r = −0.454; *p* = 0.04) and ∆ of ATG5 (r = 0.583; *p* = 0.007). Furthermore, ∆ of sNOX2-dp correlated with ∆ of ATG5 (r = −0.608; *p* = 0.004) and ∆ of NO bioavaiability (r = 0.672; *p* = 0.003).

## 4. Discussion

This study provides the first evidence that the intake of a mixture of trehalose, spermidine, nicotinamide, and polyphenols by PAD patients is associated with a significant improvement in MWD. This effect may be related to a down-regulation of NOX2-mediated oxidative stress, or to an improvement of autophagy and of endothelial function. These findings support the hypothesis that increased oxidative stress and impaired autophagy induce endothelial dysfunction and promote atherosclerotic complications in PAD patients. 

Previous findings showed that patients with PAD have severe endothelial dysfunction and reduced endothelium-mediated vasoreactivity [32] associated with both the severity and the extent of atherosclerosis in the arteries of the lower limbs [33,34]. Endothelial dysfunction is an early atherogenic event and is also associated with decreased vulnerability of plaque to rupture, potentially leading to catastrophic events.

Endothelial function can be reflected by endothelin-1 (ET-1) and nitric oxide (NO) production by endothelial cells [35]. In fact, NO released by endothelial cells induces vascular relaxation contributing to the regulation of vascular tone. Endothelial-derived NO inhibits both the synthesis and hemodynamic effects of endothelin (ET)-1 [36]. Thus, an imbalance between NO and ET-1 promotes the pathological process of the vascular diseases at their primary stage, by inducing a decrease in antithrombotic activity, as well as an increase in vascular tone and permeability and in cell growth activity. In accordance with these findings, the PAD population has increased ET1 levels [37] and lower nitrite and nitrate [38], a measure of NO, with a decreased FMD [39]. 

Elevated oxidative stress could be the pathogenic mechanism inducing the arterial dysfunction observed in PAD. Indeed, PAD patients were shown to have elevated systemic levels of several biomarkers of oxidative stress such as 8-Hydroxy-2-deoxy-2-deoxyguanosine (8-OHdG) [40], malondyaldheide (MDA), and isoprostanes [41]. In addition, PAD patients displayed increased oxidative stress, as indicated by a higher level of sNOX2dp, a marker of NOX2 activation [39].

The increase in oxidative stress impairs autophagy. Proper autophagy is an important response to the extrinsic and intrinsic cellular environments, and positively regulates cellular processes during chronic inflammation [42]. In endothelial cells, a suppressed autophagy level may contribute to further injury in human atherosclerotic lesions. Previous studies indicated that a decreased cell autophagic level was detected in ox-LDL- or TNF-α-treated HUVECs, and the induction of cell autophagy contributed to ameliorating vascular endothelial inflammation and lipid accumulation in vascular smooth muscle cells [43]. In our study, we demonstrated that autophagy is impaired in PAD patients as plasma autophagic markers ATG5 and P62 changed significantly. ATG5, a key autophagy factor, is significantly increased, whereas P62, a selective autophagy substrate, is decreased. 

Given this evidence, the supplementation with antioxidants and autophagy activators such as trehalose may be a viable countermeasure. Much evidence supports the role of polyphenols as strong antioxidants and their preventive effects against cardiovascular disease, aging, and the risk of obesity and diabetes. Among polyphenols, cocoa is a polyphenol-rich nutrient with a high content of polyphenols in beans, especially catechin and epicatechin, which were previously shown to reduce endothelial dysfunction. Indeed, previous studies demonstrated that PAD patients treated with dark chocolate, rich in epicatechin and catechin, showed a decrease in oxidative stress 2 h after its consumption [25]. 

Trehalose was also shown to be a promising agent for the prevention and treatment of a number of common health problems including cardiovascular diseases [18]. As an important autophagy modulator, it can be proficiently used in the control of several diseases in which autophagy plays a beneficial role. 

In our study, we explored the hypothesis that a mixture of trehalose, spermidine, nicotinamide, and polyphenols can improve the clinical outcome of PAD patients by positively modulating oxidative stress and autophagy. We observed that, after 60 days of intervention with the studied mixture, PAD patients displayed reduced oxidative stress biomarkers, such as NOX2dp and H_2_O_2_, compared with control patients. The beneficial effect of this mixture on oxidative status is corroborated by the increased antioxidant status, as indicated by a significant increase in the percentage of HBA, confirming that the treatment restores a balanced redox status. Moreover, the mixture can restore autophagy, as indicated by increased ATG5 and decreased P62 levels. These changes are consistent with previous study showing that a combination of polyphenols and trehalose inhibits NOX2-derived oxidative stress and significantly activates autophagy [23]. 

Finally, these changes result in the improvement in endothelial function parameters. In fact, we found that ET1 was significantly decreased and NO was significantly increased after mixture intake in PAD patients. All of these biological changes resulted in increased WMD, as demonstrated by the correlation between oxidative stress and autophagy parameters with PAD claudication. Moreover, we found a positive effect of mixture intake on diastolic pressure that decreases significantly after supplementation. These data allowed us to speculate that the mixture, by improving vascular function, reduces blood pressure, confirming that endothelial dysfunction and hypertension are integrally related [44]. 

The present study has limitations and implications that should be acknowledged. This study should be considered a proof-of-concept study that is potentially useful to understand the mechanism of PAD-related disease, and it provided evidence that needs to be confirmed in larger populations. Moreover, the study is also limited by its single-blinded design. Finally, for measuring MWD, we did not perform the treadmill test; however, several pieces of available evidence demonstrate that walking performance measured by the 6 min walk test is a reliable test–retest to assess the natural history of declines in walking endurance and detects improved walking endurance in response to therapeutic interventions in PAD patients [45].

## 5. Conclusions

Overall, the results of this study suggest that chronic administration of a mixture of trehalose, spermidine, nicotinamide, and polyphenols improves MWD with a mechanism mediated by oxidative stress and autophagy, which ultimately leads to enhanced endothelial parameters and functional improvement. Confirmation of the effect in a larger population is necessary to assess whether this mixture may be a novel approach to broaden the range of therapies for the treatment of PAD patients.

## Figures and Tables

**Figure 1 antioxidants-11-01836-f001:**
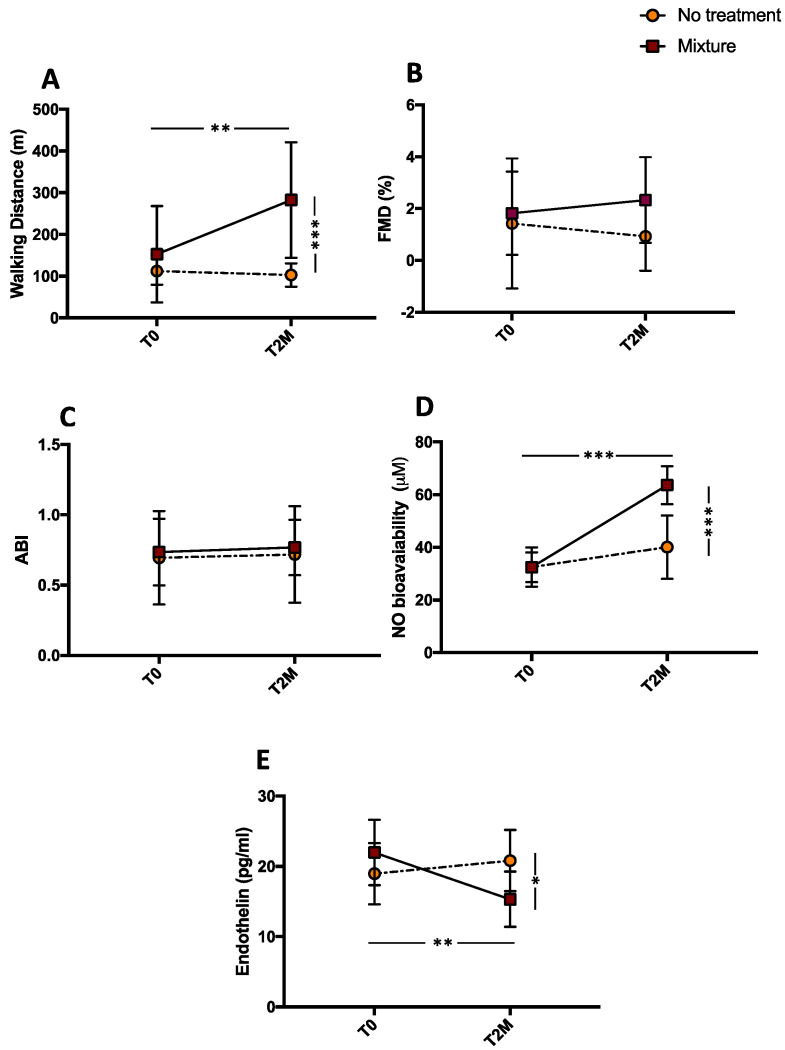
Maximal walking distance (MWD) (**A**), FMD (**B**), ABI (**C**), serum nitrite/nitrate (**D**), and endothelin-1 (**E**) before (T0) and 2 months (T2M) after mixture intake (*n* = 10) or no-treatment (*n* = 10) in peripheral artery disease patients. Data are expressed as mean ± SD. * *p* < 0.05, ** *p* < 0.01, *** *p* < 0.001.

**Figure 2 antioxidants-11-01836-f002:**
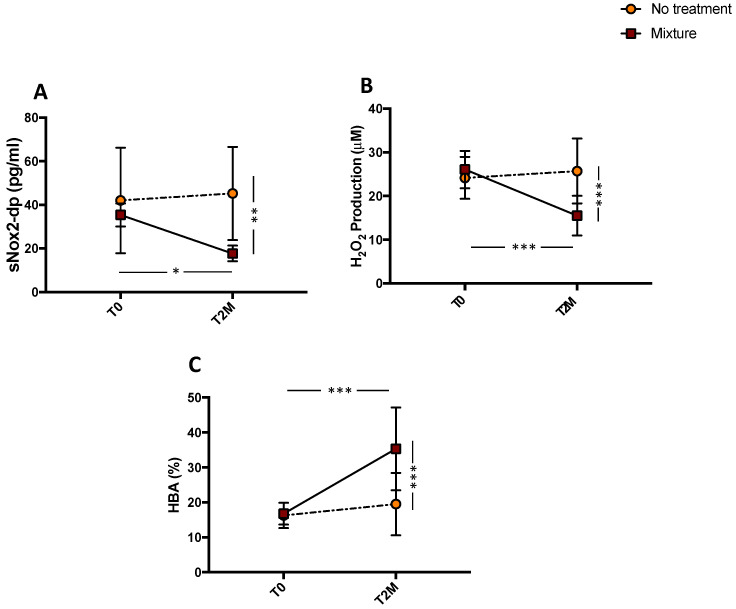
Serum soluble NOX2-derived peptide (sNOX2-dp) (**A**), serum H_2_O_2_ (**B**), and blood HBA (**C**) before (T0) and 2 months (T2M) after mixture intake (*n* = 10) or no-treatment (*n* = 10) in peripheral artery disease patients. Data are expressed as mean ± SD. * *p* < 0.05, ** *p* < 0.01, *** *p* < 0.001.

**Figure 3 antioxidants-11-01836-f003:**
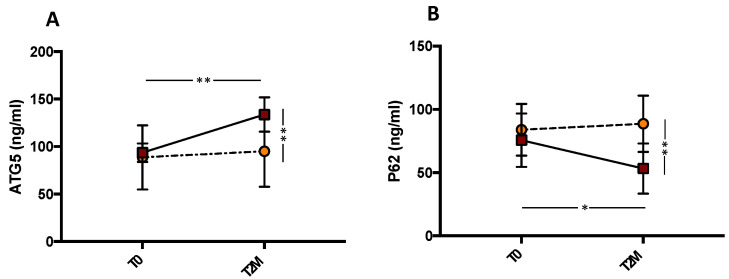
Serum ATG5 (**A**) and P62 (**B**) before (T0) and 2 months (T2M) after mixture intake (*n* = 10) or no treatment (*n* = 10) in peripheral artery disease patients. Data are expressed as mean ± SD. * *p* < 0.05, ** *p* < 0.01.

**Table 1 antioxidants-11-01836-t001:** Mixture composition.

Mixture Composition	Grams (g)
Trehalose	7.5
Spermidine	0.00225
Camellia Sinensis e.s.	0.075
Catechins	0.0375
Vitamin C	0.06
Niacin	0.00375
Silica	0.225
Microcrystalline cellulose	0.075
Orange aroma	0.3
Erythritol	2.189
Sucralose	0.036

**Table 2 antioxidants-11-01836-t002:** Clinical characteristics of PAD patients.

Variables	PAD Treatment (*n* = 10)	PAD No-Treatment (*n* = 10)
Mean age, y *	69.6 ± 8.2	75.6 ± 7.6
Males/females	7/3	6/4
Hypertension, % (*n*)	90% (9)	90% (9)
Diabetes mellitus, % (*n*)	30% (3)	40% (4)
Dyslipidemia, % (*n*)	80% (8)	50% (5)
Former smokers, % (*n*)	40% (4)	20% (2)
CHD	30% (3)	50% (5)
Previous stroke	10% (1)	0% (0)
BMI *	26.1 ± 3	25.2 ± 3
Pharmacological treatments, % (*n*)		
ACE-inhibitors	50% (5)	100% (10)
Statin	90% (9)	100% (10)
Antiplatelets	100% (10)	100% (10)
Oral anticoagulants	0% (0)	0% (0)

ACE, angiotensin-converting enzyme; BMI, body mass index; CHD, coronary heart disease; PAD, peripheral artery disease. * Data are expressed as mean ± SD.

## Data Availability

The data presented in this study are available upon request from the corresponding author.

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
