# Peer review of "Natural Activators of Autophagy Increase Maximal Walking Distance and Reduce Oxidative Stress in Patients with Peripheral Artery Disease: A Pilot Study"

_antioxidants, 2022, doi:10.3390/antiox11091836_

Round 1
Reviewer 1 Report
This study investigated the effects of treatment with a mixture of trehalose, spermidine, nicotinamide and polyphenols patients with PAD. The primary outcome measured was performance in the 6-minute walk. Measures of oxidant stress and autophagy were also reported. Some minor issues with reporting of methods and data presentation should be addressed.
Although another paper has been published, could more details of the FMD measurements be included in the methods? Was this FMD in the brachial artery or the popliteal artery?
Could the data presented in Table 2 be expanded and separated into patients given the mixture and patients who did not receive treatment? Could heart rate at rest and following the 6MWT be presented for before and after the treatment period be included? Similarly, could resting blood pressure before and after the treatment period be included?
In the figure legends, please provide separate “n” values for the no treatment and mixture groups.
In figures 2 and 3, what do the question mark symbols “?” represent?
Line 302 on page 9: What is Knox?
Line 373 on page 10: It seems as if a word might be missing after “disease related”.
Reviewer 2 Report
see enclosed word-file
